# Garlic (*Allium sativum* L.): A Brief Review of Its Antigenotoxic Effects

**DOI:** 10.3390/foods8080343

**Published:** 2019-08-13

**Authors:** José Antonio Morales-González, Eduardo Madrigal-Bujaidar, Manuel Sánchez-Gutiérrez, Jeannett A. Izquierdo-Vega, María del Carmen Valadez-Vega, Isela Álvarez-González, Ángel Morales-González, Eduardo Madrigal-Santillán

**Affiliations:** 1Escuela Superior de Medicina, Instituto Politécnico Nacional, “Unidad Casco de Santo Tomas”, Plan de San Luis y Díaz Mirón s/n, Ciudad de México 11340, Mexico; 2Escuela Nacional de Ciencias Biológicas, Instituto Politécnico Nacional, “Unidad Profesional A. López Mateos”. Av. Wilfrido Massieu. Col., Lindavista, Ciudad de México 07738, Mexico; 3Instituto de Ciencias de la Salud, Universidad Autónoma del Estado de Hidalgo, Ex-Hacienda de la Concepción, Tilcuautla, Pachuca de Soto 42080, Mexico; 4Escuela Superior de Cómputo, Instituto Politécnico Nacional, “Unidad Profesional A. López Mateos”. Av. Juan de Dios Bátiz. Col., Lindavista, Ciudad de México 07738, Mexico

**Keywords:** garlic, Ames test, sister chromatid exchange, chromosomal aberrations, micronucleus, comet assay, antigenotoxic potential

## Abstract

Traditional Medicine/Complementary and Alternative Medicine is a practice that incorporates medicine based on plants, animals, and minerals for diagnosing, treating, and preventing certain diseases, including chronic degenerative diseases such as obesity, diabetes, hypertension, atherosclerosis, and cancer. Different factors generate its continued acceptance, highlighting its diversity, easy access, low cost, and the presence of relatively few adverse effects and, importantly, a high possibility of discovering antigenotoxic agents. In this regard, it is known that the use of different antigenotoxic agents is an efficient alternative to preventing human cancer and that, in general, these can act by means of a combination of various mechanisms of action and against one or various mutagens and/or carcinogens. Therefore, it is relevant to confirm its usefulness, efficacy, and its spectrum of action through different assays. With this in mind, the present manuscript has as its objective the compilation of different investigations carried out with garlic that have demonstrated its genoprotective capacity, and that have been evaluated by means of five of the most outstanding tests (Ames test, sister chromatid exchange, chromosomal aberrations, micronucleus, and comet assay). Thus, we intend to provide information and bibliographic support to investigators in order for them to broaden their studies on the antigenotoxic spectrum of action of this perennial plant.

## 1. Introduction

Diverse evidence has demonstrated that plants perform an important role in the care and improvement of human health. The World Health Organization (WHO) estimates that more than 80% of the Earth’s inhabitants depend on or have used Traditional Medicine/Complementary and Alternative Medicine (TCAM) for their primary health care needs, conceptualizing TCAM as any practice, knowledge, and belief on health that incorporates medicine based on plants, animals, and/or minerals, spiritual therapies, manual techniques, and exercises applied individually or in combination to maintain well-being, in addition to treating, diagnosing, and preventing certain diseases, including chronic degenerative diseases (such as obesity, diabetes, hypertension, atherosclerosis, chronic respiratory diseases, and cancer) [1,2,3].

TCAM is widely employed in developed as well as in developing countries. Different positive factors generate its continuous acceptance, highlighting its diversity, flexibility, easy access, relatively low cost and, importantly, the presence of relatively low adverse toxic effects in comparison with allopathic medicine where these effects are frequently attributed to synthetic drugs. In addition, different pharmacological studies have demonstrated that there is a high possibility of discovering novel anticancerigenous, antimutagenic, and antioxidant agents from plants. Thus, in recent decades, natural antioxidants have attracted attention for their protective effects against the toxicity induced by different physical and chemical agents, and to regulate and/or treat some disorders related to the generation of free radicals present in chronic degenerative diseases. Specifically, it is known that the use of different antimutagenic and anticarcinogenic substances (in conjunction, also denominated antigenotoxic agents, to reduce the damage to genetic material) in daily life might prevent human cancer [2,3,4]. In general, antimutagenic agents have been classified as desmutagens and bio-antimutagens; the first group considers substances that promote the elimination of genotoxic agents from the organism and/or that inactivate the mutagens partially or totally by means of enzymatic or chemical interaction prior to attacking the DNA. On the other hand, bio-antimutagens can suppress the process of mutation after the DNA is damaged, as well as acting in the repair and replication processes of the damaged DNA. The mechanisms of action of the antigenotoxic agents are complex and can be classified according to the site of action or by the specific type of action (Table 1) [5,6,7]. It is important to recall that the antigenotoxic agents can act through one of a combination of several mechanisms, as well as against one or several mutagens and/or carcinogens. That is, they can present a different spectrum of antigenotoxic action. Therefore, it is relevant to evaluate their usefulness, efficacy, and spectrum of action by means of different assays.

At present, there are diverse assays, in vitro and in vivo, to determine the genoprotective capacity of the compounds; thus, it would be complicated to describe each one in detail. Generally, it may be mentioned that each test has its advantages and disadvantages, but the principal quality sought is that it be a sensitive, fast, and simple method—one capable of evaluating the genotoxic and antigenotoxic effect on somatic and germinal cells. Among the most outstanding tests of recent decades, we find the bacterial mutation assay (Ames test), sister chromatid exchange (SCE), the evaluation of chromosomal aberrations (ChAb), micronucleus assay (MN) and, more recently, the comet assay (CA) [3,6].

There are innumerable studies, it being impossible to mention all of them, and their purpose is to demonstrate and evaluate the protective and/or beneficial potential of plants, more specifically, the identification of compounds that can protect humans against damage to the genetic material generated by diverse genotoxic agents. For this reason, there are continuous efforts worldwide to explore the rich biodiversity of fruits, foods, and plants (edible and/or medicinal) in the search for more effective phyto-antimutagens. Bearing this in mind, the present report has as its objective the compiling of a great number of data based on works carried out with garlic that have demonstrated its antigenotoxic capacity, and that have been evaluated, by means of five of the most utilized models in genetic toxicology (salmonella mutagenicity test, SCE, ChAb, MN, and CA). With this objective in mind, the authors of this document intend to provide information and bibliographic support to investigators in order for them to broaden their studies on the genoprotective spectrum of this perennial plant.

## 2. Garlic

### 2.1. Overview

Garlic (*Allium sativum* L.) is a bulbous perennial plant with a powerful onion that is characterized by its peculiar aroma and pungent taste that grows up to 1.2 m in height. Depending on the geographical area where it is grown and/or used, it is also known as rocambole, allium, stinking rose, rustic treacle, nectar of the gods, camphor of the poor, poor man’s treacle, and clove garlic. *A. sativum* is the most widely consumed bulb after onion [8], it is easy to grow, and can grow in temperate and tropical regions all over the world. There are different types or subspecies of garlic, the most common being hardneck garlic and softneck garlic. Its exact origin is unknown, but it is believed to be native to Central Asia and northeastern Iran. *A. sativum* was domesticated long ago and is mentioned in ancient Egyptian, Greek, Indian, and Chinese writings. Currently, nearly 10 million tons of garlic are produced each year, with China, Korea, India, the U.S., Spain, Egypt, and Turkey the world’s largest producers [8]. In the specific case of Mexico, it is also considered an important crop, the highest production of which is centered in the states of Zacatecas, Guanajuato, and Baja California, where over 90% of the domestic production is obtained [9,10]. In the culinary field, garlic has been used as a flavoring agent and common seasoning. However, its medicinal and therapeutic benefits have been amply recognized for many years by numerous health authorities [11,12].

Its use as a therapeutic agent in TCAM extends back over 4000 years. Among the first pieces of evidence, we find 22 formulations described in the Egyptian papyrus of the Ebers Codex, which mentions it as an efficient remedy for cardiac problems, cephalea, and snake bites. In ancient Greece, garlic was consumed to treat intestinal and pulmonary disorders, while in the Second World War, it was employed as an antiseptic for the wounds and ulcers of the soldiers. In general, it has been attributed antimicrobial, antiprotozoal, antifungal, antithrombotic, analgesic, coagulant, antiasthmatic, antipyretic, antihypertensive, anticoagulant, antiasthmatic, antioxidant, and anticarcinogenic capacities [11,12,13,14].

In terms of its chemical composition, fresh garlic generally contains water, fiber, lipids, proteins, carbohydrates (such as fructose), vitamins (mainly C and A), minerals (such as potassium, phosphorus, magnesium, sodium, iron, and calcium), phytosterols, and substances of phenolic origin, as well as different organic compounds of sulfur. However, on considering solubility, its chemical composition can be divided into two large groups: (a) the first, comprising compounds of lipid-soluble allyl sulfur (such as diallyl sulfur (DAS), diallyl disulfide (DADS), and diallyl trisulfide (DATS)), and (b) water-soluble compounds, such as of g-glutamyl S-allylcysteine (SAC) and S-allyllmercaptocysteine (SAMC).

When the bulbs of fresh garlic are cut or macerated, the parenchyma is destroyed, releasing its content of S-allylcysteinesulfoxide (known as alliin). The latter is an odorless compound that is activated by the enzyme allynase to produce diallylthiosulfinate (allicin). Allicin is responsible for the odor and characteristic taste of garlic and is the main component of recently triturated garlic homogenates. However, it reacts spontaneously and can convert into more stable compounds, such as polysulfurs, among which DAS, DADS, DATS, and ajoene are prominent (Figure 1) [11,12,13,14,15].

It is known that after ingesting fresh garlic, the gastric juices start the process of transformation to different bioactive compounds, which together with the rest of the partially digested garlic are directed to the small intestine for greater decomposition and to be absorbed into the organism, whence thereafter they are distributed by the blood stream to different places, including the liver, kidneys, plasma and muscles. Some studies indicate that eating fresh garlic in excess can produce digestive disorders such as stomach acidity, indigestion, nausea, vomiting, body odor, diarrhea and gas formation. (The latter is related to the interaction of colon bacteria with the sulfur present in its bioactive compounds). Additionally, some of these compounds may irritate the digestive tract, producing a burning sensation in the abdomen and chest [16,17,18]. Generally, digestive disorders can be reduced by means of temperature, since if garlic is cooked at very high temperatures (up to 215 °C for 60 min), some of its phytocompounds can be significantly diminished. Moreover, the complete formation of allicin mediated by alliinase takes place after a period between 2 and 5 min when water is added to fresh garlic or garlic powder; however, its formation in the body after consuming is questionable, because the enzyme is inactive at a pH lower than 3.5, the range in which you commonly find the stomach, although a moderate food and/or protein-high food can increase the pH up to 4.4 where the enzyme is active [19,20]. This is why, currently, there are complementary preparations with enteric coating that include capsules or tablets to reduce these discomforts and/or to favor the administration of some of their bioactive compounds (especially the water-soluble compounds) parenteral via [21].

Thanks to its therapeutic properties and its chemical composition, garlic is considered a very versatile product in the pharmaceutical, food, and cosmetic market; therefore, different products can be produced, which have been classified into five groups as follows: (a) consumption of fresh garlic, (b) essential garlic oil, (c) macerated garlic oil, (d) garlic powder and (e) aged garlic extract [14,15].

### 2.2. Antigenotoxic Studies of Garlic

On finalizing the scientific search, employing the main electronic databases (PubMEd, SciELO, Latindex, Redalyc, BiologyBrowser, ScienceResearch, ScienceDirect, World Wide Science, Scopus, and Google) and considering the previously mentioned five experimental models in vivo as well as in vitro (Keywords: Ames test, sister chromatid exchange, chromosomal aberrations, micronucleus and comet assay), the result showed that there are approximately 31 documents related to the antigenotoxic capacity of garlic. It is curious that, despite its initial consumption from centuries ago, and that the scientific contributions on its therapeutic and/or pharmacological properties are extensive, studies on its antigenotoxic effect began in the 1990s. The first scientific contributions were directed toward the evaluation of its capacity to reduce the frequency of chromosomal aberrations and its antimutagenic potential employing the bacterial mutation assay.

In the case of the chromosomal aberrations test (ChAb), the first study found in the literature was conducted by Das et al. (1993) [22], who evaluated the anticlastogenic activity of three doses (25, 50, and 100 mg/kg) of a crude extract of garlic (CEG) against the damage produced by mitomycin C (MMC) and cyclophosphamide (CP) in mouse bone marrow cells. The results of the authors indicated that the garlic extract solely induced a low level of chromosomal damage. In contrast, when the three doses of the extract were combined with each mutagen, a significant reduction was observed in the number of ChAb, this reduction being different for each toxic agent, also confirming that this anticlastogenic effect was dose dependent.

Four years later, this same group of investigators analyzed the capacity of the same CEG (dose corresponding to 6 g for a human weighing 60 kg) to diminish the frequency of ChAb induced by sodium arsenite for a period of 30 and 60 days. Similar to the previous results, the genoprotection was significant when the extract was administered daily [23].

With the purpose of corroborating the previous results, Shukla and Taneja (2002) analyzed the antimutagenic effect of another garlic extract (GE) on newly combining it with CP. Initially, these authors carried out a treatment with 1.0%, 2.5%, and 5.0% of GE for 5 days prior to the administration of the clastogen, and 24 and 48 h afterward, the chromosomal damage was evaluated in bone marrow samples. At the end of the study, it was observed a significant decrease in cytogenetic damage with the two high doses. Additionally, it was confirmed that the GE, by itself, did not induce aberrations in either the two evaluation schedules [24].

Possibly, these three previous studies were the reason for broadening and analyzing the antigenotoxic spectrum of action of the CEG against other toxic agents, as was the case of the cisplatin and chrysotile asbestos fibers (CAFs). In the former case, it was confirmed again that a previous treatments with three different doses of CEG (125, 250, and 500 mg/kg) were capable of reducing the rate of chromosomal aberrations in germinal cells and in aberrant spermatozoa of Swiss albino mice, and as, in the results of Shukla and Taneja (2002), it was observed that this extract was not an inducer agent of ChAb. While for chrysotile asbestos fibers (CAFs), a phenomenon was observed of similar protection after simultaneously administering garlic extract (in two different concentrations: 2.0 and 5.0 μL/mL) with CAF (50 μg/mL) to a culture of human peripheral blood lymphocytes [25,26]. In sum, the conjunction of these studies suggested the following: (a) CGE was not a clastogenic agent (different result from that observed in the nineties by Das and collaborators); (b) the chemopreventive potential of CEG against chromosomal mutations can be dose dependent (with wide dose intervals); and (c) it is probable that this potential can be attributed to the interaction of the different phytocomponents of the extract.

In relation to the studies performed with the Ames test, the first contributions were developed by U.S. investigators (1997, 1999), who evaluated the formation of heterocyclic aromatic amines (2-amino-1-methyl-6-phenylimidazo[4,5-b]pyridine (PhIP) and 2-amino-3,8-dimethylimidazo[4,5-f]quinoxaline (MeIQx)) in chicken and beef meat marinated and not marinated with different substances, including garlic, brown sugar, olive oil, cider vinegar, mustard, lemon juice with salt, Teriyaki sauce, curcumin sauce, and barbecue sauce with commercial honey. After submitting the two meat types to the process of cooking on a grill, the levels were determined of both heterocyclic aromatic amines (HAA) by liquid–liquid extraction and gas chromatography–mass spectrometry, and the possible antimutagenic potential of the meat extracts utilizing the bacterial mutation assay. In sum, the results showed that the marinating process (especially, with garlic) reduced the total of detectable HAA and their mutagenic activity, observing that this diminution could be modify on terms of cooking time. These data suggested to the investigators that a prior marinade with garlic can reduce the concentration of these mutagens (especially, PhIP amine) formed in the cooking process [27,28].

The results of both studies may call attention to the fact that the process of biotransformation of fresh, ground or powder garlic (often used for marinating) can be affected by very high cooking temperatures, which would modify the presence of some bioactive protector compounds. Therefore, it would be important to analyze and consider the temperature, humidity, type of garlic used and the time of ripening. In this sense, there is some evidence that indicates: (a) It is considered that between 160 and 200 °C for 30 min, there is still stability in organosulfur compounds; (b) some foodstuffs cooked with garlic and/or acidified that do not have alliinase activity have shown a higher allicin bioavailability than expected, for example, boiled garlic (16%), baked (30%), marinated (19%) and chopped with acid (66%); (c) It is also known that aged garlic (AGE) is processed and ripened under a controlled temperature and humidity for over one month, and contains a larger amount of S-Allylcysteine (SAC) than fresh garlic, so if powdered AGE was used in previous studies, the observed bioprotective effect was possibly attributed to the presence of SAC [19,20,21].

Later, Ikken et al. (1999) analyzed the antimutagenic effect of nine ethanolic extracts of fruits and vegetables against N-nitrosamines (N-nitrosodimethylamine (NDMA), N-nitrosopyrrolidine (NPYR), N-nitrosodibutylamine (NDBA), and N-nitrosopiperidine (NPIP)). The extracts of kiwi, onion, and garlic were those that showed the greatest antimutagenic effect (in the range of 50–2000 microg/plate), especially against NDBA, NPIP, and NPYR [29].

In the most recent scientific contribution carried out to date (2006), the purpose was to evaluate the antimutagenic and immunomodulator potential of some substances employed frequently in the human diet. Using the same test as that in prior studies, the genoprotective property was confirmed of various homogenates derived from cauliflower, garlic, onion, red cabbage, carrot, and broccoli against the mutagenicity induced by aflatoxin B_1_ (AFB_1_), 2-amino-3-methylimidazo [4,5,-f] chinolin (IQ) and N-nitroso-N-metylurea (MNU). It was also observed that the homogenates demonstrated a clear reduction of the immunosuppression exercised by the mutagens, with garlic one of the most representative [30].

The majority of scientific evidence on the antigenotoxic capacity of *Allium sativum* has been conducted employing the micronucleus assay (MN). Approximately 14 studies have been developed from the year 1995, all of which have demonstrated that garlic can be an ideal candidate for consideration as a good anticlastogenic agent. In general, these contributions are evaluations in vivo (there are practically only two studies in vitro) against exposure to various mutagens and/or carcinogens (Table 2). The agents tested were physical and chemical, and included the following: gamma radiation, N-methyl-N′-nitro-N-nitrosoguanidine (MNNG), sodium arsenite, chrysotile asbestos, cyclophosphamide (CP), 7,12-dimethylbenz [a] anthracene (DMBA), 2-acetyl aminofluorene (2-AAF), N-methyl-N′-nitro-N-nitrosoguanidine, aflatoxin B_1_ (AFB_1_), 2-amino-3-methylimidazo [4,5,-f] chinoline (IQ) and N-nitroso-N -methylurea (MNU), ethylenediaminetetraacetic acid (EDTA) and cypermethrin (CYP). In sum, the majority of the results conclude that garlic extracts are not cytotoxic or genotoxic agents (although there are some data that suggest a slight increase in MN); contrariwise, they reduce the frequency of the MN [especially micronucleated polychromatic erythrocytes (MNPE)]. Similarly, the data suggest that their genoprotective effect is dose dependent and can be associated with their antioxidant property.

With respect to the sister chromatid exchange (SCE) test, described by Perry and Wolff in 1974, the basis of which consists of the observation of chromosomes with chemically differentiated chromatids on employing a differential stain in which an analog of thymine (5-bromodeoxyuridine (BrDU)) was incorporated for two cell cycles [43]. To our knowledge, there are only two studies on the genoprotective capacity of *Allium sativum* in the literature. This can be attributed to the fact that, in recent years, this evaluation test has been replaced by techniques that are more innovative and faster to carry out, in that, to obtain and observe the SCE, it is indispensable to employ BrDU and Fluorochrome Hoescht 33,258 and have access to ultraviolet light. In general, SCE are induced by a great variety of agents, whether they are those that interfere directly or indirectly with the process of replication or those that produce breaks in the chains of genetic material, with the exception of restriction enzymes and drugs that act as anti-toposomerases. Additionally, it has been observed that the genotoxic agents that act during phase S of the cellular cycle are good inducers of SCE and of chromosomal aberrations [43,44]; thus, in the following studies, both parameters will be analyzed. The former was already mentioned in the paragraph on chromosomal aberrations and refers to the result obtained in the culture of peripheral blood lymphocytes in which the diminution was evaluated of the genotoxicity induced by chrysotile asbestos using garlic extract (GE), observing that the extract, at two different concentrations (5.0 and 10 µg/mL), was capable of reducing the rate of SCE as well as that of chromosomal aberrations [20]. Likewise, Sowjanya et al. (2009) confirmed this capacity of GE, but in this instance against DNA damage induced by CP in a similar experimental model [45], with which both studies reaffirm the suggestion of the authors that, in the future, investigations should be conducted on the design and development of possible modulator drugs that contain garlic extract. After the MN test, the comet assay (CA) (also known as the single cell gel electrophoresis (SCGE) assay or the microgel electrophoresis (MGE)), is the second model of evaluation with an important number of studies related to the genoprotective potential of garlic. The CA was introduced by Östling and Johanson [46] to detect DNA damage induced by radiation. However, the alkaline method, developed by Singh et al. [47] which allows the denaturalization of the DNA, as well as the detection of alkaline-marked sites, has become the most utilized and recommended assay due to its broad detection spectrum of DNA damage. This technique has grown exponentially, especially in the new millennium. It is considered a simple, sensitive, and rapid methodology for detecting simple- and double-chain DNA breaks, alkaline-labile sites, DNA oxidative damage, DNA cross-linked outcomes, DNA adducts, apoptosis, and necrosis [48]. On average, there are seven scientific contributions that have been developed since 2006, being the same Singh and collaborators who initiated this exploration. In this study, the authors investigated the protector efficiency of three garlic powders administered to rats for 2 weeks against DNA damage induced by N-nitrosodimethylamine (NDMA) and AFB_1_. At the end of the established period, these authors confirmed that the powders reduced between 35 and 60% the genetic damage produced by both mutagens in the liver and colon of the animals. With these data, the authors considered that garlic deriving from fields fertilized with sulfur probably have a greater amount of allin, a sulfur phytochemical, to which this protective efficacy can be attributed [49]. The rest of the investigations are presented in Table 3; in sum, animals were employed in the five studies, (mainly mice), and different cellular cultures were utilized (such as leukocytes, human amniotic cells, human lung cells, and Vero cells) in the others. Similarly, these investigations confirmed that garlic (in the form of powders, extracts, homogenates, and oil) notably reduces the frequency of single- or double-strand breaks of DNA produced by the hydrogen peroxide (H_2_O_2_), N-nitrosodimethylamine (NDMA), 2-amino-3-methylimidazo [4,5,-f] chinoline (IQ), N-nitroso-N -methylurea (MNU), some mycotoxins (AFB_1_ and zearalenone (ZEN)), an antiviral agent employed in patients with HIV (stavudine (Zerit, d4T)), environmental compounds that generate ecological problems (such as lead acetate (Led) and mercury chloride (Mer)) and tributyltin (TBT), a biocide that can be transmitted to humans by contaminated shellfish.

## 3. Perspectives and Conclusions

The investigations shown in the present review demonstrate the usefulness of the five assays ((Ames test, sister chromatid exchange (SCE), chromosomal aberrations (ChAb), micronucleus (MN), and comet assay (CA)) to confirm the antigenotoxic effects of garlic (*Allium sativum* L.) against different mutagenic and/or carcinogenic agents both in vivo and in vitro. In this regard, it is important to consider that there is diverse evidence of the action of the physical, chemical, and biological agents on the genetic material, resulting in mutations that can be associated with genomic instability and the development of cancer [48].

Therefore, the main regulatory agencies, such as the Federal Food and Drug Agency (FDA), the European Medicines Agency (EMA), and the Agencia Nacional de Vigilancia Sanitaria (ANVISA, Brazil), suggest genotoxicity/antigenotoxicity tests as an essential part of the validation of the drugs and/or substances destined for human consumption [48]. These tests include in vitro and in vivo assays to detect the potential of the compound to induce and/or reduce genetic mutations and/or chromosomal aberrations. Guideline S2 (R1) on Genotoxicity Testing and Data Interpretation for Pharmaceuticals Intended for Human Use (generally applied by these agencies) suggests two options of batteries of tests: In the first option, there are assays in vitro, such as the Ames test, the genetic mutation test in mouse lymphoma TK cells, and/or the evaluation of chromosomal aberrations and MN. In the second option, the inverse mutation test in bacteria is again considered, and the genotoxic evaluation in vivo in two tissues is incorporated mainly hematopoietic tissues (using MN assay), and the other, in which the CA is evaluated [48].

However, this guideline allows the use of different methods in order for the investigators to be able to determine the safety and/or genoprotection of the compounds. As can be observed, CA as well as MN are prominent among the evaluation tests, possibly due to their properties of robustness sensitivity, rapidity, and statistical power to evaluate DNA breaks, which can be considered characteristics of mutagenicity [48,55]. In addition, current studies point out that the association between CA and MN is a good option for evaluating the mutagenic and antimutagenic potential, in that they permit the detection of breaks in the chromatic and chromosomal levels, respectively [48]. This is possibly the reason for which, in this document, a greater number of antigenotoxic studies carried out with these assays is observed (14 for MN and seven with CA).

With respect to the mechanism(s) of antigenotoxic action of garlic, the majority of the studies presented in this manuscript suggest that the most frequent mechanism is related to its antioxidant activity. However, the authors of this review, as well as other investigators, conclude that there might be more mechanisms involved. In this regard, various modes of action have been proposed considering its antigenotoxic and anticarcinogenic potential. Among these, the following are included: (a) the effect on the metabolizer enzymes of drugs (that is, the induction of enzymes of disintoxication in phase II of including glutathione transferase, quinine reductase, epoxide hydrolase, and gluconosyl transferase, which inactivate toxic substances and facilitate their excretion [12,56]); (b) the inhibition of tumor growth, which has been documented in several cell culture lines of carcinoma, including prostate carcinoma cells [12,56]; (c) the induction of apoptosis, which coincides with an increase in the percentage of cells blocked in the G2/M phase of the cellular cycle [12,56]; (d) the effective stimulation of the immune system (a phenomenon observed mainly in the organosulfur compounds of their chemical composition, which stimulate the proliferation of lymphocytes and the phagocytosis of macrophages, and they stimulate the release of interleukin-2 (IL-2), tumor necrosis factor α (TNF-α), and interferon gamma) [12,56].

Likewise, the majority of studies on its genoprotective potential conclude and suggest that this property can be attributed to the interaction of the different phytocomponents present in its chemical composition, which opens the possibility of exploring a new scientific document on the analysis of this potential considering its principal phytochemicals (Allicin, ajoene, DAS, DADS, DATS, SAC, and SAMC). However, it would be important to consider the process of obtaining each garlic extract, which is relevant to the presence of each phytochemical group, and therefore on the observed genoprotective effect. In general, the extracts shown herein were aqueous (obtained from distilled and deionized water), ethanolic and methanolic. In some cases, garlic oil as well as ground and dried powder were used.

As mentioned in the Introduction, antigenotoxic agents can act against one or several mutagens of carcinogens. The results of the investigations presented suggest that the antigenotoxic spectrum of garlic is broad and can be dose dependent; in addition, it appears that it can be considered a safe agent in its different presentations of use; however, this is not totally definitive and to fully confirm its protector spectrum, it would be convenient to conduct more studies employing other mutagens and/or carcinogens of direct and indirect action, to evaluate other dosage intervals, and to indisputably broaden the investigations on its toxic potential, in that, to date, there are contradictory data on this aspect (that is, some evidence indicates that, depending on the dose and/or the time of exposure administered, it could induce mutations, chromosomal aberrations, and MN).

On the other hand, despite the fact that the consumption of garlic is frequent and that there are diverse scientific contributions on its therapeutic properties, pharmacological, and anticarcinogenic properties, it would be important to extend the scientific investigations on its antigenotoxic effects in controlled clinical studies in humans employing the MN assay and/or the comet assay (CA). In this context, it is convenient to consider that antigenotoxic markers can be relevant in the prediction and/or development of cancer, the latter having been suggested on correlating it with some epidemiological evidence in which it has been considered that it is possible to prevent cancer and other chronic diseases—some of the latter sharing common pathogenic mechanisms, such as DNA damage, oxidative stress, and chronic inflammation [57,58,59,60]. However, the positive results of the antigenotoxic agents observed in preclinical studies are not always the same when they are analyzed in clinical assays, in which variable data have been obtained. That is, on occasion, similar positive results have been demonstrated, but at different percentages, while unfortunately, in other cases, the data are not conclusive. This suggests that more complete and solid clinical assays should be carried out [61,62,63].

Finally, it is convenient to consider that garlic organosulfur compounds have generally been considered beneficial to health, generating a frequent and excessive consumption, and for long periods, in the human population. However, there is some evidence that these phytocompounds may lead to toxicity and adverse effects, which suggests possible dual biological functions. In fact, they could act as biological double-bladed swords, for it has been observed that organosulfur compounds (such as thioles and disulfides) are redox cyclers. In aerobic conditions, the thiols experience a spontaneous oxidation to disulfides that, in turn, easily reduce to thiols. During this redox cycle, thio radicals and reactive oxygen species are produced which include hydrogen peroxide, superoxide radicals and hydroxyl radicals. Likewise, it has been found that ajoene, diallyl sulfide, diallyl disulfide, diallyl trisulfide, allylmethyl sulfide, allyl methyl trisulfide and dipropyl sulfide may show a pro-oxidative potential, hepatocarcinogenic, cytotoxic, mutagenic and/or be SCE inducers [19,64].

Taking all of the previously mentioned material together, it is clear that it would be convenient to continue more investigations (both preclinical and clinical) on this perennial plant to confirm its safety and to fully determine its antigenotoxic capacity and, thus, use garlic in health for its chemopreventive properties.

## Figures and Tables

**Figure 1 foods-08-00343-f001:**
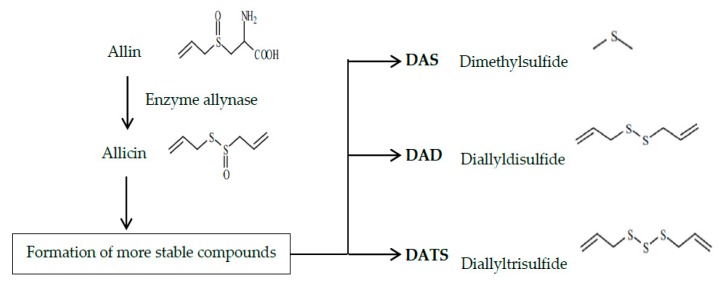
Allin transformation to more stable compounds.

**Table 1 foods-08-00343-t001:** Main mechanisms of antigenotoxic action.

Action Site	Types of Mechanisms
Extracellular	Inhibition of mutagen uptake
	Inhibition of endogenous formation (a) Inhibition of nitrosation (b) Modification of the intestinal flora
	Complexation and/or deactivation
	Favouring absorption of protective agents
Intracellular	Blocking or competition (a) Scavenging of reactive oxygen species (b) Protection of DNA nucleophilic sites
	Stimulation of trapping and detoxification in non-target cells
	Modification of transmembrane transport
	Modulation of xenobiotic metabolising enzymes (a) Inhibition of promutagen activation (b) Induction of detoxification pathways (c) Inhibition of metabolic enzymes
	Modulation of DNA metabolism and repair
	Regulation of signaling pathways
	Enhancement of apoptosis
	Maintenance of genomic stability

Table modified from Ferguson et al. (2004) [3,5].

**Table 2 foods-08-00343-t002:** Studies carried out with the micronucleus (MN) assay that evaluate the antigenotoxic effect of garlic.

Year	Aim of the Study	Conclusion	Type of Study	References
1995 1996	Singh et al. evaluated the genoprotector effect of a garlic extract (GE) against the chromosomal damage induced by three doses (0.5, 1.0, and 2.0 Gy) of gamma radiation. Previously, three doses of GE were administered orally (125, 250, and 500 mg/kg) for 5 days and, later, the mice were irradiated.	The results demonstrated that the previous treatment significantly reduced the frequency of micronucleated polychromatic erythrocytes (MNPE) induced by radiation and that the protector effect was dose dependent.	In vivo	[31,32]
2001	The objective of the study was to determine the inhibitor effect of GE on the clastogenicity of N-methyl-N′-nitro-N-nitrosoguanidine (MNNG) in Wistar rats. Similar to the two studies by Singh et al., the GE was previously administered (250 mg/kg) orally and, later, the mutagen was injected.	At the end of the study, it was observed that the pretreatment of the aqueous extract diminished the number of extracts induced by the nitrosamine carcinogen.	In vivo	[33]
2003	In this study, in vivo, the modulating potential of four aqueous extracts (Garlic, ginger (*Zingiber officinale*), sconio (*Pimpinella anisumm* LINNE), and cloves (*Syzygium aromaticum*)) was evaluated on the clastogenic effects of sodium arsenite in rat bone marrow cells.	The results of the investigation confirmed the following: (a) All of the extracts showed slight clastogenicity on the order of garlic > ginger, and sconio > cloves; (b) The previous treatment for 7 days with the extracts of the condiments diminished the formation of MNPE induced by the inorganic clastogen, being on the order of ginger > garlic > cloves > sconio and (c) This anticlastogenic effect can be attributed to the antioxidant properties of their chemical components.	In vivo	[34]
2004 2006	Considering that chrysotile asbestos fibers (CAFs) are environmental carcinogens, researchers from India evaluated the reduction of their genotoxicity in human peripheral blood lymphocytes by means of a garlic extract (GE). Additionally, through the electronic spin resonance (ESR), they determined the generation of hydroxyl radicals.	Their results showed that, when it was administered to the cellular culture, simultaneously the asbestos (1.0 µg/mL) plus the extract at two different concentrations (5.0 and 10 µg/mL), a clear reduction was observed in the induction of MN. On the other hand, the ESR test demonstrated that the asbestos alone induces nearly two times the signals of ESR, which diminished with the presence of the GE. The conjunction of these data suggested that the protector effect of the extract can be related to its antioxidant potential.	In vitro	[26,35]
2004	This study analyzed the interactive effects of saffron with garlic and curcumin on the genotoxicity induced by cyclophosphamide (CP) in mouse bone marrow. The animals were orally treated previously with saffron (100 mg/kg), garlic (250 mg/kg), and curcumin (10 mg/kg), alone or in combination, for 5 days, 2 h prior to the administration of the CP.	The maximal reduction of the frequency of MNPE induced by the mutagen was observed when the three test compounds were administered together. In addition, the protective effects were more pronounced in the group treated with garlic in comparison with the groups administered with curcumin and/or saffron.	In vivo	[36]
2004	The objective of the investigation was to determine the combined effect of orally administration tomato and garlic in the face of the genetic damage induced by 7,12-dimethylbenz [a] anthracene (DMBA) in Swiss albino mice. Different experimental groups were included, among these those treated with each extract individually and the combination of both compounds (tomato (500 mg/kg) and garlic (125 mg/kg)) plus the administration of the DMBA. In addition to evaluating the incidence of MN, there was the determination of lipid peroxidation and the concentrations of glutathione, glutathione peroxidase, and glutathione-S-transferase in the liver	On finalizing the experiment, it was confirmed that the animals treated with DMBA presented a greater frequency of MN and an elevated lipid peroxidation. Although the previous treatment with tomato or garlic significantly reduced genotoxicity and the parameters of oxidative stress induced by DMBA, the combination of both extracts showed a more profound effect of this inhibition. These data suggested that the combination of functional foods (especially in doses corresponding to average daily consumption, as in the case of this experiment), can be an effective alternative for inhibiting damage to the DNA produced by carcinogens.	In vivo	[37]
2005	The protection against the toxicity of 2- acetyl aminofluorene (2-AAF) exerted by the Garlic (Ga), Honey (Ho) and Bitter kola (Bk) was investigated. Mice organized into different experimental groups were dosed for 7 days with suspensions of Ga, H, and Bk as dietetic supplements; on termination of this period, they were administered a unique intraperitoneal dose of 2-AAF. Finally, the degree of clastogenicity/anticlastogenicity and liver damage was measured.	The results revealed the carcinogen induced the rate of MNPE of the bone marrow by approximately five times in comparison the control group; contrariwise, on being combined with the different suspensions of the dietetic supplements, the capacity of 2-AAF diminished to induce MN on the order of Ho > Ga > Bi. Likewise, it was observed that the activity of the gamma glutamyltransferase induced by the carcinogen was reduced on the order of Ho > Bi > Ga (serum) and Bi > Ga = Ho (liver).	In vivo	[38]
2005	The purpose of the study was to evaluate the attenuation of the genotoxicity and of the oxidative stress produced by the MNNG through pretreatment with garlic and tomato in Swiss mice using the MN assay and estimating the extension of lipid peroxidation and the state of the antioxidants of the redox cycle of the glutathione.	At the conclusion of the study, the following were confirmed: (a) the increase in MN and of lipid peroxidation is associated with the disequilibrium of the antioxidant defenses generated by the carcinogen; (b) the previous treatment with tomato and garlic (especially when administered in a combined manner) attenuated the genotoxicity induced by MNNG, and (c) the genoprotective effect of these natural compounds was associated with the reduction of glutathione (GSH) and of the enzymes dependent on GSH, glutathione peroxidase (GPx), and glutathione-S-transferase (GST).	In vivo	[39]
2006	The purpose of the research was to examine the anticlastogenic potential of the different vegetable homogenates (including garlic, red cabbage, cauliflower, broccoli, carrot, and onion), and some chemically identified substances in pure form (resveratrol, diallylsulfide, phenethylisothiocyanate, ellagic acid, epigallocatechin gallate, genistein and curcumin) against AFB_1_, IQ and MNU.	In general, the results evidenced that all of the homogenates and the substances of plant origin showed an important anticlastogenic effect against the three mutagens. However, garlic was found among the most significant homogenates.	In vivo	[30]
2008	The study objectives included the following two: (1) to evaluate the potency of ethylendiaminotetraacetic acid (EDTA) to induce biochemical changes, bone marrow micronuclei, and damage to the DNA, and (2) to determine the inhibitory capacity of the extract of Panax ginseng and garlic on the toxic effects of the EDTA in adult male albino rats.	The group administered only with EDTA alone showed a diminution in the biochemical parameters of the serum and the activity of the antioxidant enzymes. Also increased were lipid peroxidation and the incidence of MNPE, while in animals with the combined treatment with Panax ginseng and garlic plus EDTA significantly improved all of the parameters tested.	In vivo	[40]
2010	There is clear evidence that the insecticide cypermethrin pyretroid (CYP) is a clastogenic agent. Due to this, the protector effect of an extract of garlic (500 mg/kg) and vitamin C (VTC) against the cytogenic damage induced by CYP in the bone marrow of male white rats was analyzed. Similar to other studies, GE as well as VTC were administered individually and in combination orally for 5 days prior to the application of CYP.	As expected, the administration of CYP significantly clastogenic effects, that is, it raised the frequency of MNPE and of structural chromosomal aberrations in cells in metaphase of the bone marrow. On the other hand, the results revealed the genoprotective effect of GE and VTC, especially when they are administered in combined fashion.	In vivo	[41]
2013	Although the anticancerigenous property of the garlic has been demonstrated in different studies, there is little evidence of its protector ability on the adverse effects of chemo- and radiotherapy. In order to clarify this, a mouse model was established with a tumor xenograft through a subcutaneous injection of H22 tumor cells, which were employed to investigate the genoprotective capacity of garlic oil (GO) on chemo- and radiotherapy. For the first evaluation, the animals who were tumor carriers were treated for 14 days with cyclophosphamide (CP) individually and with the combination of CP plus GO (25 and 50 mg/kg), while for the radiotherapy test, the animals were radiated only once (5 Gy) and were administered with GO.	The results showed that GO did not increase the rate of inhibition of CP/radiation of the tumor, which indicated that GO could not increase the chemo/radiosensitivity of the cancerous cells. In addition, the treatment with GO significantly inhibited the diminution of the contents of DNA and the proportion of bone marrow micronuclei. These findings support the idea that the consumption of GO can benefit patients with cancer who receive chemotherapy or radiotherapy.	In vivo	[42]

**Table 3 foods-08-00343-t003:** Studies carried out with the comet assay (CA) that evaluate the antigenotoxic effect of garlic.

Year	Aim of the Study	Conclusion	Type of Study	References
2006	Singh et al. evaluated the protective effect of three garlic powders (GP) obtained from bulbs cultivated in soils with different levels of sulfur against DNA damage induced by N-nitrosodimethylamine (NDMA) and AFB_1_	After 2 weeks of pretreatment with the GP, a reduction was observed of between 35 and 60% of the damage generated by both mutagens. This supports the idea that fertilization with sulfur can exert an impact on the genoprotection of the garlic bulbs, which is probably related to an increase in the content of the alliin, a sulfured phytochemical of garlic.	In vivo	[49]
2006	The objective was to determine the capacity of a garlic homogenate (GH) and the diallylsulfur (DAS) to reduce DNA fragmentation produced by AFB_1_, IQ, and MNU.	The results showed that GH as well as DAS fulfilled the expected objective, notably protecting the genetic material from the damage induced by the three mutagens. The latter supports the evidence that phytochemicals in the diet can perform important functions as chemopreventive agents.	In vivo	[30]
2007	In this investigation, the genomic damage was analyzed that was caused by stavudine (anti-HIV infection drug (Zerit d4T)) and the possible effect of its improvement on employing garlic oil (GO) and vitamin E (VTE). Two doses (low and high) of GO and VTE were administered, separately and in combination for 6 days, to the animals and, later, a sole dose of Zerit d4T.	VTE as well as GO, separately and in combination, reduced the clastogenicity of Zerit d4T, observing that the genoprotective effect was more pronounced with the high dose of GO plus the vitamin. These results suggest that both agents work interactively by means of an antioxidant mechanism.	In vivo	[50]
2007	Tributyltin (TBT) is a biocide employed as an additive in anti-fouling paint on the hulls of ships, wharfs, and buoys, to avoid the growth of marine organisms. Unfortunately, it can be transmitted to humans by contaminated shellfish and, to date, no effective strategy is known to eliminate the toxic effects of these foods. Therefore, in this investigation, the capacity of garlic oil (GO) to prevent the damage produced by TBT in vivo and in vitro was explored.	It was observed that, in mice as well as in amniotic cell cultures (human FL), when treated previously with GO, this significantly diminished the production of reactive oxygen species and the number of DNA cells damaged by TBT. This suggested that GO can reduce the oxidative damage induced by TBT in vivo as well as in vitro through an antioxidant mechanism.	In vivo In vitro	[51]
2009	Considering that there is evidence that high temperatures and/or cooking processes can affect the bioactivities of fruits and vegetables, the antioxidant and antigenotoxic effects of an Aged garlic extract (AGE) were analyzed in comparison with a raw garlic extract (RGE) and a heated garlic extract (HGE) obtained by different processing methods	At the end of the experiment, the results showed the following: (a) the greatest total phenol content corresponded to AGE; (b) the antioxidant activity evaluated through DPPH indicated that the HGE was significantly higher, while the activity of the SOD was in decreasing order RGE > AGE > HE, and (c) AGE was the extract type that notably reduced the damage to the DNA induced by H_2_O_2_ in the culture of leukocytes, showing an inhibition rate of approximately 70%. The data suggested that the thermal process can diminish the antioxidant and genoprotector activity of the garlic.	In vitro	[52]
2012	The objective of the present study was to evaluate the protective capacity of an aqueous extract of *Allium sativum* (AEA) against the cytotoxicity induced by zearalenone (ZEN), the generation of reactive oxygen species (ROS), and the fragmentation of the DNA in a culture of Vero cells. In general, the cytotoxicity was analyzed utilizing the MTT viability assay, while the antioxidant activity was carried out measuring the activity of the catalase. Finally, to determine whether the induction of oxidative stress was associated with the DNA lesions, the DNA fragmentation was sought by means of the comet assay (CA).	The results indicated that ZEN induced several toxic effects and significant alterations measured by its action in the oxidative stress, while in the combined treatment of ZEN plus AEA (250 μg/mL), an important reduction was observed of all of these damages in all of the markers evaluated, especially a significant reduction of DNA fragmentation generated by the mycotoxin	In vitro	[53]
2018	Evaluation of the interactive effect of the garlic and the vitamin E (VTE) against the cytotoxic and genotoxic damage of lead acetate (Led) and the mercury chloride (Mer) in a culture of human pulmonary cells (WI-38). Initially, the WI-38 cells were treated with garlic and VTE for 24 h and, later, with Led and/or Mer alone or combined for 24 h.	In general, it was observed that Led or Mer or the combination of both induced serious damage to the DNA, a phenomenon that was reverted when the cells were pretreated with VTE plus garlic. The latter suggests that the garlic can interact with the vitamin, generating a very promising protector effect against the toxic effect of heavy metals.	In vitro	[54]

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
