# Peer review of "Garlic (Allium sativum L.): A Brief Review of Its Antigenotoxic Effects"

_foods, 2019, doi:10.3390/foods8080343_

Round 1

Reviewer 1 Report

The paper presents a comprehensive review of the anti-genotoxic properties of garlic, summarising the findings of the most relevant in vitro and in vivo studies. The authors conclude that for a thorough proof that garlic can be used in chemoprevention further studies to prove its efficacy are required.

The information is presented in a well-organized manner, focused on the genotoxicity assays used to demonstrate the anti-genotoxicity of the garlic extract. The references cited along the text are credible and the article is written in a correct English, which contributes to its good readability.

Thus, I recommend this article for publication in Foods after addressing some minor issues, as follows:

Line 33 - To avoid repetition please replace the expression “With this objective in mind” by, e.g., “Thus”

Lines 58-61 – I do not completely agree that “the use of different antimutagenic and anticarcinogenic substances (…) in daily life is an efficient procedure for preventing human cancer and some genetic diseases.” I suggest replacing “is an efficient procedure for preventing” by “might prevent”. In addition, it is not clear to me which genetic diseases can be prevented - please provide some examples.

Lines 124 – 137 – I suggest the inclusion of the chemical structure of (some of) the garlic bioactive compounds referred in this piece of text.

Line 139 and following – As this is a review paper, can the authors detail further how the literature search was performed e.g., which sources were used, the key words selected?

Were the garlic extracts mentioned along this review obtained by the same method? Please add some information about this along the text because different extraction procedures may lead to different extract composition, which might influence the effect observed.

Lines 163-164 – please reword “a decrease in dose-dependent cytogenetic damage”

Line 168 and following – please add the animal species used

Lines 225 – 232 – please replace “chemically different chromatids” by “chemically differentiated chromatids”; “been displaced by techniques” by “been replaced by techniques” and delete the word “microscope” from the sentence “have access to an ultraviolet light microscope” because only the UV light is needed and the preparations are analysed under a bright field microscope.

Lines 310 – 319 –the authors state that despite the recognized antioxidant activity, garlic extracts present other beneficial properties. To support this please add references at the end of each item (from a) to d)).

Line 337 - please specify what type of studies you are referring to. Clinical trials?

Line 351 – Finally, garlic cannot be called a chemopreventive agent. It is better to say that garlic can be used for its chemopreventive properties. Please rephrase the last sentence accordingly.

Author Response

Title: Garlic (Allium sativum L.): a brief review of its antigenotoxic effects

Manuscript ID: Foods-545097

Reviewer 1

Review Report (Round 1)

Comments and Suggestions for Authors

The paper presents a comprehensive review of the anti-genotoxic properties of garlic, summarising the findings of the most relevant in vitro and in vivo studies. The authors conclude that for a thorough proof that garlic can be used in chemoprevention further studies to prove its efficacy are required.

The information is presented in a well-organized manner, focused on the genotoxicity assays used to demonstrate the anti-genotoxicity of the garlic extract. The references cited along the text are credible and the article is written in a correct English, which contributes to its good readability.

Thus, I recommend this article for publication in Foods after addressing some minor issues, as follows:

Line 33 - To avoid repetition please replace the expression “With this objective in mind” by, e.g., “Thus”

Lines 58-61 – I do not completely agree that “the use of different antimutagenic and anticarcinogenic substances (…) in daily life is an efficient procedure for preventing human cancer and some genetic diseases.” I suggest replacing “is an efficient procedure for preventing” by “might prevent”. In addition, it is not clear to me which genetic diseases can be prevented - please provide some examples.

Lines 124 – 137 – I suggest the inclusion of the chemical structure of (some of) the garlic bioactive compounds referred in this piece of text.

Line 139 and following – As this is a review paper, can the authors detail further how the literature search was performed e.g., which sources were used, the key words selected?

Were the garlic extracts mentioned along this review obtained by the same method? Please add some information about this along the text because different extraction procedures may lead to different extract composition, which might influence the effect observed.

Lines 163-164 – please reword “a decrease in dose-dependent cytogenetic damage”

Line 168 and following – please add the animal species used

Lines 225 – 232 – please replace “chemically different chromatids” by “chemically differentiated chromatids”; “been displaced by techniques” by “been replaced by techniques” and delete the word “microscope” from the sentence “have access to an ultraviolet light microscope” because only the UV light is needed and the preparations are analysed under a bright field microscope.

Lines 310 – 319 –the authors state that despite the recognized antioxidant activity, garlic extracts present other beneficial properties. To support this please add references at the end of each item (from a) to d)).

Line 337 - please specify what type of studies you are referring to. Clinical trials?

Line 351 – Finally, garlic cannot be called a chemopreventive agent. It is better to say that garlic can be used for its chemopreventive properties. Please rephrase the last sentence accordingly.

Answers

Dear reviewer

The authors appreciate the comments and observations of the article.

Thanks for everything

Receive a cordial greeting

Line 33. The observation is correct. We have considered the observation and include the word “Thus”

Lines 58-61. Now is line 60. The observation is correct. We have modified the wording of the paragraph and include the phrase "could prevent"

Lines 124-137. Now is line 134. We have considered the observation and include figure 1

Line 139 and following. Now 143-147. We have considered the observation and include the keywords and the main electronic databases that were used

Now 334-338. We also consider the second comment. We include in the Perspectives and conclusion section a brief paragraph that mentions the relevance of the extraction process and the types of extracts mentioned in the document

Lines 163-164. Now 169-170. We have considered the observation. We have modified the wording of the paragraph.

Line 168 and following. Now 178. We have considered the observation. We have included the strain of mice used in the study.

Lines 225-232. Now 234-241. We have considered all observations and/or comments.

Lines 310-319. Now 323-329 The observation is correct. We have considered the observation and include the references in each ítem.

Line 337. Now 351. We have considered the observation. We modified the wording and now it is mentioned: “clinical studies in humans”

Line 351. Now 364-365. We have considered the observation. We have modified the wording of the paragraph.

Reviewer 2 Report

I have reviewed an article: "Garlic (Allium sativum L.): a brief review of its 3 antigenotoxic effects". It is a review paper including chronologically report of potential proctective action of biologically active compounds present in garlic.

There are some issues I would like to ask authors to respond to;

1) Please indicate in the text what is the stability of biologically active compounds during cooking (any kind of thermal processing), since in one part of the text you have mentioned that garlic can prevent formation of pyrolitic compounds during grilling.

2) Please add a paragraph concerning stability of bioactive compounds from garlic during absorption and possible metabolism  conducted by intestinal microflora.

What intermediates are formed during these processes and what is their potential genotoxic/protective nature?

3) Are biologically active compounds from garlic metabolized in liver? If so, what intermediates are formed? Also, indicate potential protective/genotoxic potential of formed intermediates.

4) Are parent compounds distributed through organism?

Author Response

Title: Garlic (Allium sativum L.): a brief review of its antigenotoxic effects

Manuscript ID: foods-545097

Reviewer 2

Review Report (Round 1)

Comments and Suggestions for Authors

I have reviewed an article: "Garlic (Allium sativum L.): a brief review of its 3 antigenotoxic effects". It is a review paper including chronologically report of potential protective action of biologically active compounds present in garlic.

There are some issues I would like to ask authors to respond to;

1) Please indicate in the text what is the stability of biologically active compounds during cooking (any kind of thermal processing), since in one part of the text you have mentioned that garlic can prevent formation of pyrolitic compounds during grilling.

2) Please add a paragraph concerning stability of bioactive compounds from garlic during absorption and possible metabolism  conducted by intestinal microflora.

What intermediates are formed during these processes and what is their potential genotoxic/protective nature?

3) Are biologically active compounds from garlic metabolized in liver? If so, what intermediates are formed? Also, indicate potential protective/genotoxic potential of formed intermediates.

4) Are parent compounds distributed through organism?

Answers

Dear reviewer

The authors appreciate the comments and observations of the article.

Thanks for everything

Receive a cordial greeting

Observation 1). Lines 218-229. We appreciate the comment and observation. We have considered this suggestion and include a paragraph that mentions the effect of temperature on the transformation process of fresh, crushed or powdered garlic (types of garlic generally used to marinate). In this same context, we have mentioned information about the possible stability of the metabolites and which of the metabolites could be involved in the protective effect that the authors of the article have suggested.

Observation 2) Lines 134-152. We appreciate the comment and observation. We have considered this suggestion and include a paragraph that mentions brief information about the pharmacokinetics of garlic and its metabolites. Likewise, we have mentioned some factors that can affect this process. In this same context, we have mentioned brief information about the use of some garlic supplements (different pharmaceutical presentations) that are used to favor its administration by other routes.

Observation 2 and 3) Lines 390-401. We appreciate the comment and observation. We have considered this suggestion and include a paragraph that mention: Finally, it is convenient to consider that garlic organosulfur compounds have generally been considered beneficial to health, generating in the human population a frequent and excessive consumption, and for long periods. However, there are some evidences that these phytocompounds may lead toxicity and adverse effects, which suggests possible dual biological functions. In fact, they could act as biological double bladed swords, for it has been observed that organosulfur compounds (such as thioles and disulfides) are redox cyclers. In aerobic conditions, the thiols experience a spontaneous oxidation to disulfides that, in turn, easily reduce to thiols. During this redox cycle, thio radicals and reactive oxygen species are produced which include hydrogen peroxide, superoxide radicals and hydroxyl radicals. Likewise, it has been found that ajoene, diallyl sulfide, diallyl disulfide, diallyl trisulfide, allylmethyl sulfide, allyl methyl trisulfide and dipropyl sulfide may show a pro-oxidative potential, hepatocarcinogenic, cytotoxic, mutagenic and / or be SCE inducers.

In addition, in the same section "Perspectives and conclusion" we mentioned that it would be convenient to write a related document on garlic metabolites, explaining about their genotoxic and antigenotoxic effects. This information is relevant and extensive. Basically, the objective of this document is centered on the garlic genoprotective effect.

Observation 4) With respect to the comment or question ...Are parent compounds distributed through organism? We apologize.

We have doubts about your question. We would like to know more what the question refers to in order to respond appropriately. In this sense, we understand that the parenteral pathways are intradermal (ID), subcutaneous (SC), intravenous (IV) and intramuscular (IM) and in these cases the process of absorption and distribution are different. For example: the IV pathway would enter the bloodstream directly. In general, the studies we show in the document indicate that garlic was mainly administered orally (enteral route)
